# Correlating metasurface spectra with a generation-elimination framework

Jieting Chen[1,2,3], Chao Qian [1,2,3] ✉, Jie Zhang[1,2,3], Yuetian Jia[1,2,3] & Hongsheng Chen [1,2,3] ✉

Inferring optical response from other correlated optical response is highly demanded for vast applications such as biological imaging, material analysis, and optical characterization. This is distinguished from widely-studied forward and inverse designs, as it is boiled down to another different category, namely, spectra-to-spectra design. Whereas forward and inverse designs have been substantially explored across various physical scenarios, the spectra-to-spectra design remains elusive and challenging as it involves intractable many-to-many correspondences. Here, we first dabble in this uncharted area and propose a generation-elimination framework that can self-orient to the best output candidate. Such a framework has a strong built-in stochastically sampling capability that automatically generate diverse nominations and eliminate inferior nominations. As an example, we study terahertz metasurfaces to correlate the reflection spectra from low to high frequencies, where the inaccessible spectra are precisely forecasted without consulting structural information, reaching an accuracy of 98.77%. Moreover, an innovative dimensionality reduction approach is executed to visualize the distribution of the abstract correlated spectra data encoded in latent spaces. These results provide explicable perspectives for deep learning to parse complex physical processes, rather than "brute-force" black box, and facilitate versatile applications involving cross-wavelength information correlation.

Remarkable progress has been made in recent years in the development of intelligent metamaterials that involve deep learning to mutate the design and working mode of metamaterials[1–3]. As artificially engineered structures, metamaterials have emerged as a key player in manipulating electromagnetic (EM) waves with unparalleled optical properties. In particular, their planar equivalence, metasurfaces, have recently gained wide popularity owing to the negligible thickness, better integration, and lower insertion loss[4]. These breakthroughs have motivated scientists to revisit established physical concepts and facilitated a myriad of exciting applications that cannot be replicated with natural materials[5,6]. In these studies, a pivotal step is to design a meta-atom structure and spatiotemporal metasurface layout for specific requirements[7,8]. Although the degrees of freedom in metasurface design are tremendous and flexible, conventional bottom-up design strategy can hardly be generalized into concrete guidelines. Conventional metasurface designs mostly rely on site-specific physical models, such as equivalent circuit model[9], and lengthy full-wave numerical simulations, such as the finite-difference time domain[10]. For a given EM response, researchers often need to finely tune the geometries and iteratively search for an optimal answer in a prescribed manner. This procedure necessitates repeated manual labor and expensive

[1]ZJU-UIUC Institute, Interdisciplinary Center for Quantum Information, State Key Laboratory of Extreme Photonics and Instrumentation, Zhejiang University, 310027 Hangzhou, China. [2]ZJU-Hangzhou Global Science and Technology Innovation Center, Key Laboratory of Advanced Micro/Nano Electronic Devices & Smart Systems of Zhejiang, Zhejiang University, 310027 Hangzhou, China. [3]Jinhua Institute of Zhejiang University, Zhejiang University, 321099 Jinhua, China. ✉e-mail: chaoq@intl.zju.edu.cn; hansomchen@zju.edu.cn

computing resources, and the design result is often hampered by the existing design templates and limited physical intuition.

Data-driven methods based on deep learning allow a computational model to learn representations of data with multiple levels of abstraction and thus carry out tasks without explicit programmed and procedural instructions. Besides its mainstream applications in classification[11], translation[12,13], drug discovery[14], and neuroscience[15,16], deep learning has recently started to interact with metamaterials. Such interaction promises to significantly accelerate the design of photonic structures and circumvent low-efficiency drawbacks in conventional design strategy[17,18]. Intelligent metasurface design can be generally divided into two classes: forward[19,20] and inverse design[21]. Inverse design is opposite to forward design, meaning the direct generation of the metasurface candidates for the user-defined optical response. Yet, inverse design is not simply reciprocal to forward design due to the non-uniqueness issue. To date, we have witnessed both forward and backward designs have been well studied with different network structures, with the common goal of building up a bidirectional "expressway" between the design parameters and optical response[22–24].

However, there is another class of metasurface design that has been ignored, i.e., the inference of optical response from other correlated optical response. This is instrumental in a wide range of applications, for example, to extract the desired images, spectra, and material features from other easily accessible information with the advantages of low cost and easy measurement[25]. To distinguish it from forward and inverse designs, we term it the third-class metasurface design. Achieving third-class metasurface design faces a formidable challenge because it involves complex many-to-many mapping, rather than simply one-to-one and one-to-many mapping[26,27]. Many-to-many mapping means that there are multiple correct answers for one given input and vice versa. In doing so, conventional deep learning algorithms will become conflicted on how to adjust learnable weights and the convergence cannot be guaranteed[28,29]. Besides, the inherent bias in conventional one-to-many network cannot be evaluated and mitigated in virtue of the structural information anymore. The direct connection of two standard one-to-many networks will cause the magnification of the error. Thus, it is imperative to develop a powerful and general network architecture and facilitate the third-class metasurface design.

In this work, we for the first time dabble in this uncharted area and propose a generation-elimination framework to automatically generate many possible output candidates and then retain the optimal. Such framework consists of two cascaded networks, namely, generation and elimination networks. For a given input, the generation network is capable of producing diverse candidates by sampling over its latent space based on a variational auto-encoder (VAE) structure[30]. Then, the elimination network eliminates all inferior candidates through the merging of two latent spaces; the operation process is understandable as a hierarchical bifurcating tree, which builds up a two-way mutual selection between two spaces and eliminates the inherent error to the greatest extent. We take the terahertz metasurface as an example to demonstrate the inference of reflection spectra from low to high frequencies. After the neural networks have been trained, the retained output shows a precise match with the ground truth. Compared with the existing practices of forward and inverse designs, our generation-elimination mechanism provides a general and efficient gateway to find out the optimal solution in complex and non-unique correspondences. Such ability is important in tackling many problems involving many-to-many mapping that is commonly encountered in metasurface applications, such as invisibility cloak[31–33], optical computing[34–37], and wireless communication[38].

## Results
### Concept of the third-class metasurface design
The bidirectional non-uniqueness predicament is prevalent in metasurface and photonic design. A representative example is to infer the optical response from the low-frequency to the high-frequency band. Such inference is meaningful for avoiding expensive high-frequency detections, recovering high-frequency information after Fourier transform, and reducing meshed design space and simulation time. For example, in communication and Raman spectroscopy, in conjunction with the low-frequency correlation, the noise in the high-frequency region can be largely wiped off[39,40]. Similarly, in the spectral analysis of proteins that contain high-frequency noise, the weaknesses in smoothing and peak identification in convention methods can be overcome by virtue of low-frequency component[41]. And as a more concrete example, the low-resolution images obtained from the synthetic aperture radar (SAR) system could be transitioned to the high-definition one with negligible processing time for some real-time applications[42]. However, the realization of this spectral correlation is difficult and cannot be easily duplicated from conventional forward and inverse designs due to the complex many-to-many mapping. Currently existing designs intrinsically solve one-to-one or one-to-many mapping issues, eliding the nature of physical interpretability and the solution diversity.

Our third-class design strategy is delineated in Fig. 1. As a representative and generalizable example, we consider an elliptical-shaped metasurface, where the interested frequency band ranges from $f_1$ to $f_3$. A magnifying glass is leveraged to schematically represent our VAE-based sub-networks and its latent space. The generation "magnifier" is trained to encode low ($f_1$ to $f_2$) and high ($f_2$ to $f_3$) frequency bands into the latent space using basic gradient update and back-propagation algorithm. The optimal candidates labeled from 1 to $k$ are generated (the blue spectra in Fig. 1) by sampling over its latent space when given the input spectra (the red spectra in Fig. 1), without consulting structural information. A similar operation is also applied for another VAE-based sub-network, i.e., elimination "magnifier". Compared with conventional forward/inverse design, our third-class metasurface design heralds a novel class of metasurface design and generalizes the prevailing paradigms.

### Network architecture
Figure 2a schematically depicts the network architecture that contains two cascaded networks, namely, generation network and elimination network. As its name implies, the generation network serves as a producer to manufacture diverse candidates, and the elimination network acts as an inspector to pick out the optimal one. Both networks are composed of an encoder, a latent space, and a decoder. When two cascaded networks have been trained independently, only their decoders will be preserved and combined by a hierarchical bifurcating tree. As shown in Fig. 2b, the bidirectional mapping and two-way selection are built up between low-frequency and high-frequency spaces, and the inferior candidates can be excluded by comparing the Euclidean distance between the label input and secondary candidates in the low-frequency space. The detailed layer-level information is referred in Fig. 2c. For simplicity, only the generation network is drawn, and the elimination network is easily reproduced by swapping the locations of low and high frequencies. The reflection coefficient during the high-frequency band from 60 to 100 THz is discretized into 668 data points. Owning to the geometrical symmetry of the ellipse pattern, only three reflection coefficients, $R_{xx}$, $R_{xy}$ and $R_{yy}$, are considered, constituting a 2004-dimensional input (output) vector. Notice that in interpreting the structure of the generation network, both input and output refer to the same high-frequency component (the output $x'$ can be deemed as the reconstruction of the input $x$); this is consistent with the working mechanism of auto-encoder and VAE. Meanwhile, the "input" we mentioned in Fig. 1 corresponds to the "label" ($y$) in both Fig. 2a, c. It serves as another input element as indicated by the blue boxes. The interested region of the label starts from 40 to 60 THz and is discretized into 333 data points, constituting a 999-dimensional label vector. It is the label that transforms an

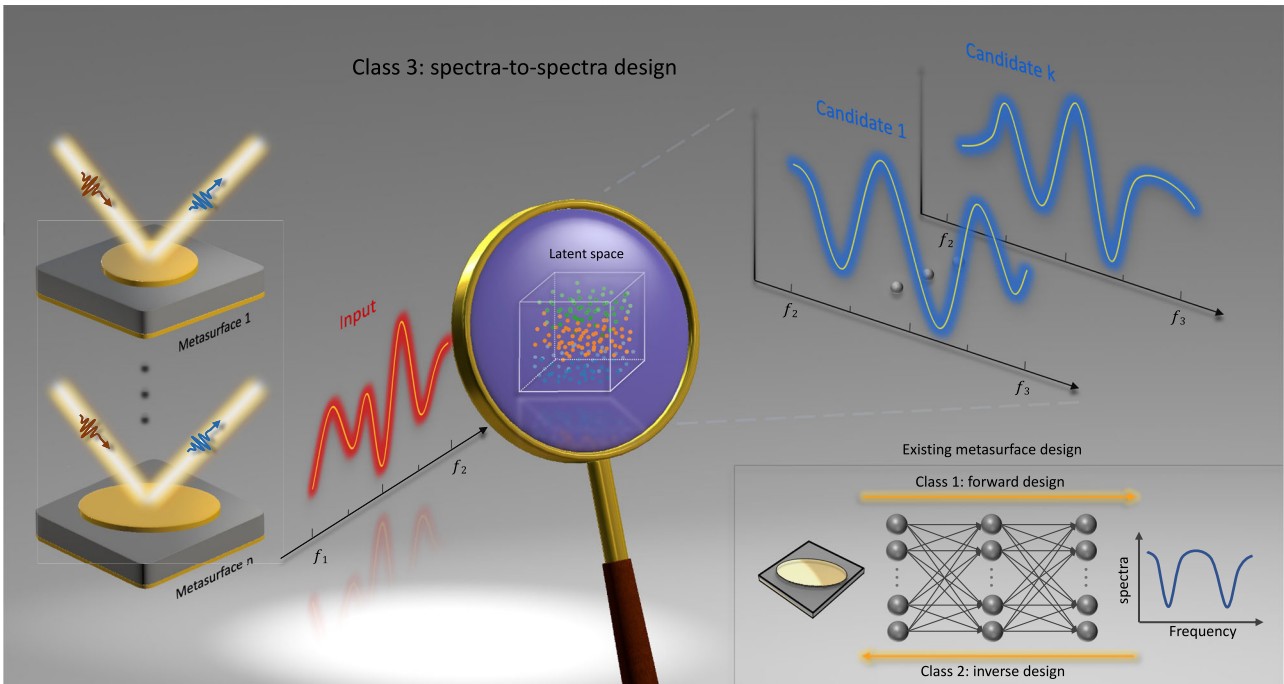

**Fig. 1 | Illustrative procedure of metasurface spectral correlation.** Despite a variety of network architectures and physical scenarios, the common practices behind the existing metasurface designs bascially regard the reciprocal transformation of physical structures and spectra as forward (one-to-one) or inverse (one-to-many) mapping process, as shown in the lower right. In our proposal, taking the inference of optical response from low-frequency to the high-frequency band as an example, we first explore the third-class metasurface design, i.e., spectra-to-spectra design. This is much challenging and non-trivial due to the bidirectional non-uniqueness predicament. To facilitate the understanding, we leverage a magnifying glass to schematically represent the VAE-based generation network and its latent space. Without consulting any structure information, the input spectra combined with stochastically sampled latent variables will produce diverse output candidates, which waits to be further picked in the elimination network.

unsupervised model VAE (Bayesian model[30,43]) into a supervised model conditional VAE (conditional Bayesian model[44]).

As indicated in Fig. 2c, there are three types of variables in our VAE-based sub-network: input or output variable $x$ (high spectra), label variable $y$ (low spectra), and latent variable $z$ (the prior probability $p_\theta(z)$ is modeled by standard Gaussian distribution $p_\theta(z) \sim \mathcal{N}(0,I)$). The graphic mathematical relationship of these three variables is discussed in Supplementary Fig. S1. For the training phase, first, a feature extraction module composed of 4 fully connected layers is used to compress the input $x$ into a lower dimension. Then, the input feature is concatenated with the label information $y$ and passes through another three concatenation layers. The mid-part of the sub-network embodies how we create the latent space in Fig. 2a, where multivariate Gaussian-based distribution $q_\varphi(z|x,y)$ is introduced to approximate the true posterior distribution of one-to-many mapping $p_\theta(z|x,y)$. Since we employ Gaussian to perform variational approximation, the variational parameter $\varphi$ (the family of distributions) would be the mean and variance of the latent variables for each data point $\varphi_{x_i} = (\mu_{x_i}, \sigma^2_{x_i})$. In our scene, considering the time cost and efficiency maximization, only two sets of Gaussian distribution variables, $(\mu_1, \sigma_1)$ and $(\mu_2, \sigma_2)$, are generated from the network neurons of the preceding layer, spawning a 2D latent space. To offer a clear-cut similarity between the approximate posterior $q_\varphi(z|x,y)$ and the true posterior $p_\theta(z|x,y)$, we define a Kullback–Leibler divergence term $KL[q_\varphi(z|x,y)||p_\theta(z|x,y)]$ to assess the information lost between the probability distribution and the approximated one. The ultimate goal is to seek out the optimal variational parameter $\varphi$ to minimize the above term. Due to the existence of intractable peaky evidence term $p_\theta(x|y)$, the divergence is hard-to-reach via direct computation; see Supplementary Note 1. Hopefully, in virtue of ELBO (i.e., the evidence lower bound, Eq. (S4)), it could be transformed into an alternative objective function, defined as:

$$\mathcal{L}_{CVAE}(x,y;\theta,\varphi) = KL[q_\varphi(z|x,y) || p_\theta(z|y)] - \mathbb{E}_{q_\varphi(z|x,y)}\left[\log p_\theta(x|z,y)\right] \quad (1)$$

where the first term is KL divergence loss, and the second term is an expected reconstruction loss. The detailed formulas of the objective function are left in Supplementary Note 1.

Two latent Gaussian variables are then sampled from the latent space and concatenated with the label information $y$ once again. After going through a reconstruction module consisting of five fully connected layers, they are finally decoded into the reconstructed output $x'$. The distance between $x$ and $x'$ is calculated as a negative maximum log-likelihood in the second term of Eq. (1). During the inference phase with user-defined or testing labels, we randomly sample two Gaussian variables from $\mathcal{N}(0,I)$, performing with the identical concatenating and decoding processes to generate realistic output candidates. For further illustration, the whole latent space is composed of countless standard Gaussian distributions with spatial continuity after training. The label seeks out a specific standard Gaussian distribution, and then the variables are randomly sampled from the selected Gaussian distribution and decoded into candidates, constituting the solution domain in Fig. 2b.

Take it into a nutshell, when training the sub-network conditioned on labels, the input and output are required to be consistent while retrieval diversity needs to be guaranteed. In other words, the reconstruction loss and KL divergence loss are always in opposition to each other until convergence, that is, a one-to-many mapping relationship is established. After that, the random sampled variables combined with the testing label can retrieve plenty of candidates in the inference phase.

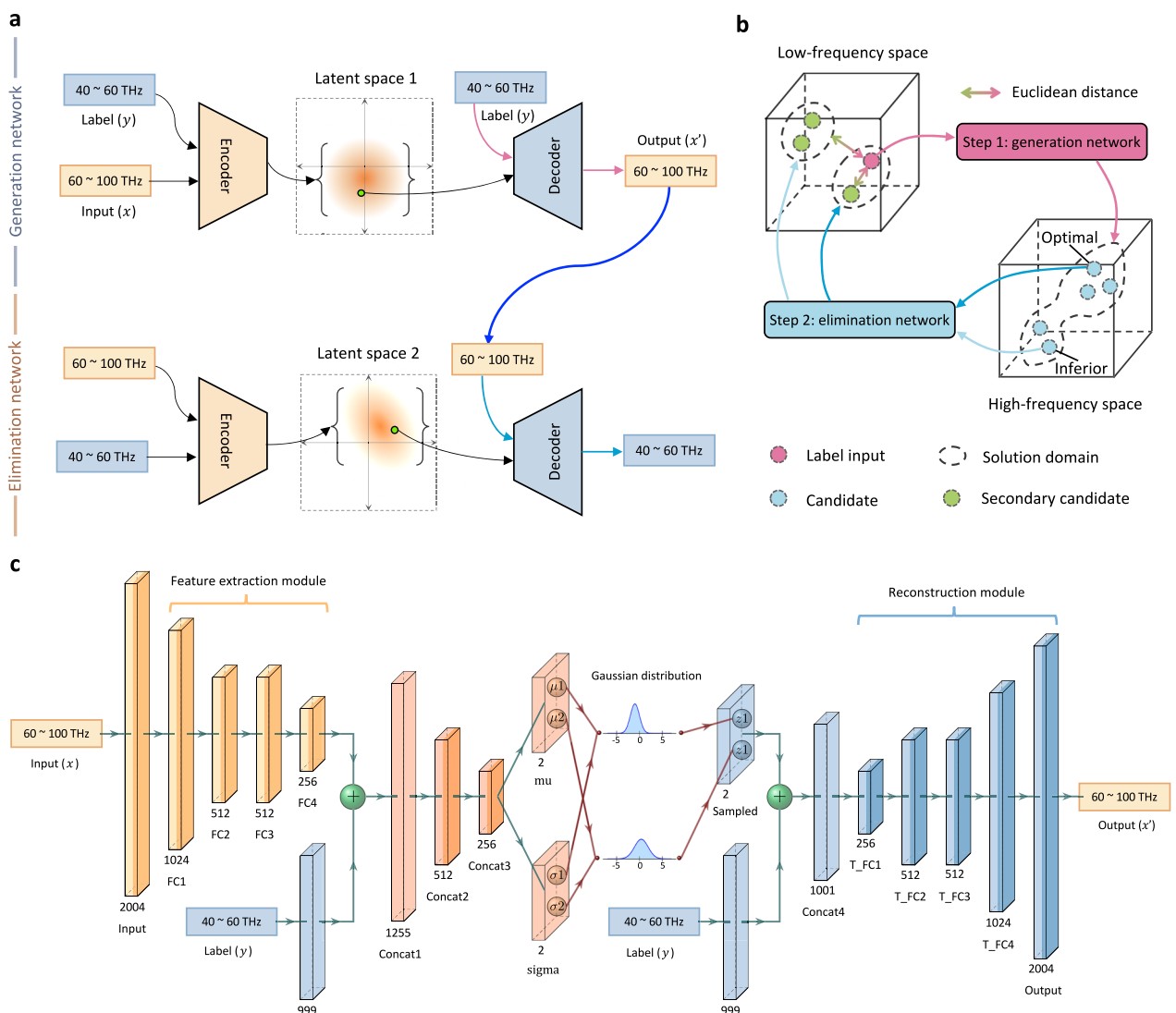

**Fig. 2 | Network architecture. a** A macro perspective to look at the architecture of the proposed network consisting of two cascaded networks, namely, generation network and elimination network, each of which is composed of an encoder, the latent space, and a decoder. **b** The interconnection between two networks. For a given input (low frequency), the generation network can generate various candidates (high frequency), and the elimination network will reversely map each candidate into the original space. The optimal candidate is picked out by calculating the Euclidean distance between the input and secondary candidates. **c** The layer-level illustration of the generation network. The feature extraction module combined with three concatenation layers composes the encoder, while the reconstruction module combined with *Concat4* composes the decoder. Sampled variables from the two sets of Gaussian distributions (i.e., variational posterior) constitute the latent space.

## Visualizing latent space with auto-encoders

To visualize the distribution of latent space 1 in Fig. 2c and verify its capability in encoding input/output data, we deploy auto-encoder approach to compress the high-dimensional spectra data into continuous one-dimensional data. Distinct from conventional methods that represent discrete categories with one-hot vectors (e.g., geometry categories in metasurface design or digital labels in handwriting recognition), our continuous spectra data are more abstract and unrepresentable. In doing so, two auto-encoders are employed to learn the compressed and distributed spectra representations (i.e., encodings), which are simplified to $e_1$ (input data) and $e_2$ (output data) ranging from 0 to 1; see the detailed auto-encoder structure in Supplementary Note 2.

Figure 3a shows the 2D distribution of 600 encoded training data featured and colored by 40–60 THz spectra representation. Whenever $e_1$ is assigned, a Gaussian distribution will be extracted (Supplementary Fig. S4), which accords with our pre-defined standard Gaussian prior distribution $p_\theta(z) \sim \mathcal{N}(0, I)$. Figure 3b shows the corresponding low-

frequency input spectra ($e_1 = 0.4$), and 11 points are circled from the extracted latent Gaussian distribution in Fig. 3a. For each point, if we concatenate these 2D latent variables with label information (i.e., the input spectra in Fig. 3b) and execute the reconstruction module (analogous to the inference phase), their corresponding high-frequency spectra will be decoded, as showcased in Fig. 3c. More broadly, any variable $z$ sampled from standard Gaussian distribution combined with label information can be decoded into a nominated solution, not merely 11 training data points.

In addition, by feeding both spectra into the sub-network, the latent variables are expected to encode the information on 60–100 THz at the same time. Figure 3d shows the selfsame 2D latent distribution of encoded training data yet featured by 60–100 THz spectra representation $e_2$. All points are clearly separated into several clusters, suggesting that our VAE-based sub-network is able to automatically distinguish different high-frequency spectral curves, without providing the corresponding spectra representation. For example, point 1 and point 2 with

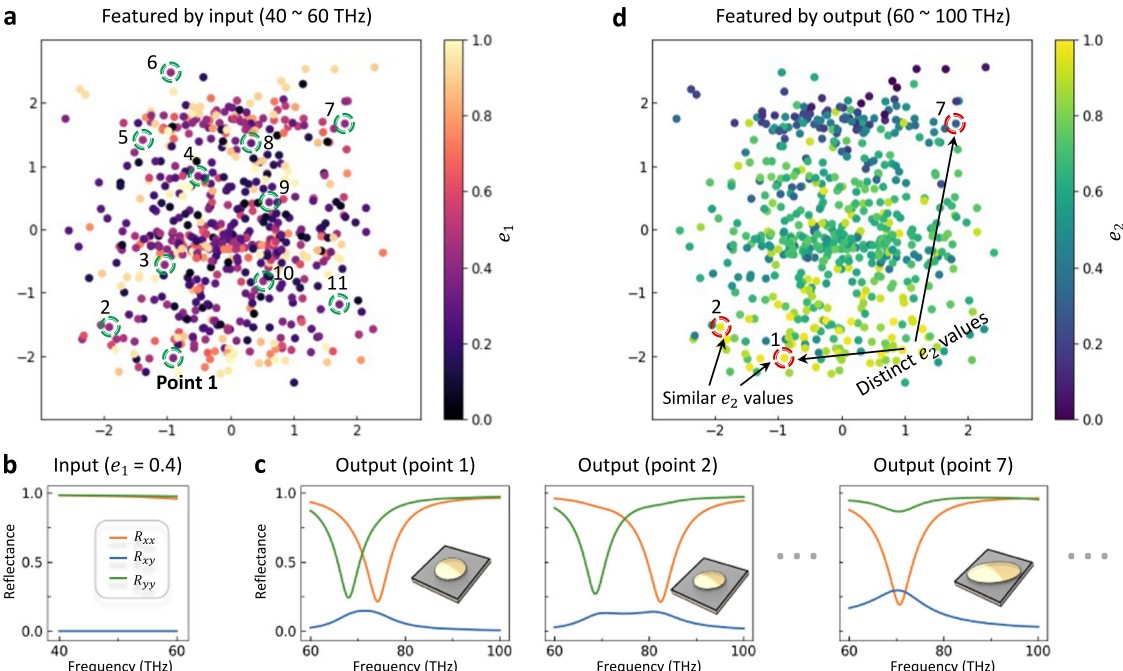

**Fig. 3 | Feature extraction and latent space visualization. a** 2D distribution of 600 encoded training data featured and colored by 40–60 THz input spectra representation. The 999-dimensional inputs (i.e., labels) are compressed into 1-dimensional spectra representations ($e_1$) with the help of the auto-encoder, whose value ranges from 0 (pale yellow) to 1 (dark violet). 11 points are circled from the extracted latent Gaussian distribution with $e_1 \approx 0.4$. **b** The corresponding low-frequency input spectra when $e_1 = 0.4$. **c** The ground-truth spectra of 11 points and their metasurface designs. For each point, the 60–100 THz output spectra are retrieved by concatenating the 2D latent variable with the 999-dimensional input (i.e., label information) and then going through the reconstruction module. **d** The same 2D distribution of 600 encoded training data featured and colored by 60–100 THz output spectra representation. The 2004-dimensional outputs are compressed into 1-dimensional spectra representations ($e_2$) with another trained auto-encoder. All points are clearly separated into different clusters, where the points possessing similar $e_2$ values tend to have similar spectrum curves (such as, point 1 and point 2) and vice versa.

similar $e_2$ values will have nearly the same spectral curves (as shown in Fig. 3c), whereas the spectrum curves of point 1 and point 7 are quite divergent because their $e_2$ values are different. This not only validates the effectiveness in dimensionality reduction by auto-encoder, but also proves that the latent variables with only two dimensions can conditionally encode input and output spectra into a compact but informative space.

### Nominated results from the generation network

To confirm the superiority of our generation network, we compare our method with others (Fig. 4). Three samples are blindly selected from the shuffled testing dataset and their input spectra from 40 to 60 THz are plotted in Fig. 4a. We first evaluate the performance of conventional fully connected network. By treating third-class metasurface design as a biunique problem and just using several fully connected layers, conventional fully connected network (FCN) arouses distinct inconsistencies between the ground truth (the solid lines) and predicted spectra curves (the dashed lines), as shown in Fig. 4b, exposing its shortcoming of ignoring the non-convergence nature in the third-class metasurface design; see Supplementary Note 2 for more details. On the contrary, our generation network produces diverse and more precise candidates in Fig. 4c. By sampling latent variables from the prior standard Gaussian distribution, we obtain retrieved candidates colored with different transparencies in dashed lines, all of which are nominated results waiting to be further filtered in the elimination network. Here, for the sake of brevity, we draw only three sets of candidate solutions, yet at least one of which is in good agreement with the ground truth. It verifies that our generation network is superior in solving such non-convergence problems.

### Optimal results from elimination network

An elimination network is indispensable by virtue of two aspects. First, the generation network embodies a one-to-many mapping from low to high-frequency band. Accordingly, an inverse one-to-many mapping of optical response from high to low-frequency band is required. In other words, every optimal 60–100 THz nomination will generate various 40–60 THz secondary candidates, constituting a data structure analogous to the tree diagram in Fig. 5a. Second, the KL divergence term in Eq. 1 reveals that there is a deviation between the true latent distribution and standard Gaussian prior distribution. It means that the latent variables, though sampled from the acceptable region, are not guaranteed to retrieve highly precise nominations. As delineated in Fig. 2a, the elimination network has a similar structure to the generation network. The only difference is that the label variable $y$ (input or output variable $x$) is assigned with the high-frequency (low-frequency) spectra. It is worth emphasizing that two sub-networks share the same training dataset, without any glance at the testing dataset during the training phase.

Furthermore, we exploit a tree diagram to clarify how we combine the generation "magnifier" and elimination "magnifier" to build up the bidirectional mapping between two spectra spaces in Fig. 2b and obtain the ideal solution. As depicted in Fig. 5a, several father nodes discovered by the generation "magnifier" are linked to the input root node. Then, each father node gives birth to some secondary candidates termed as leaf nodes by using the elimination network. The core is that the father node with the most orthodox leaf node will be chosen as the final output, and "the most orthodox" is defined as the minimum distance between the root node and the leaf node, or alternately say, the minimum Euclidean distance between the input and secondary candidates in Fig. 2b. The procedure is somehow similar to the "cycle consistency" criterion in the

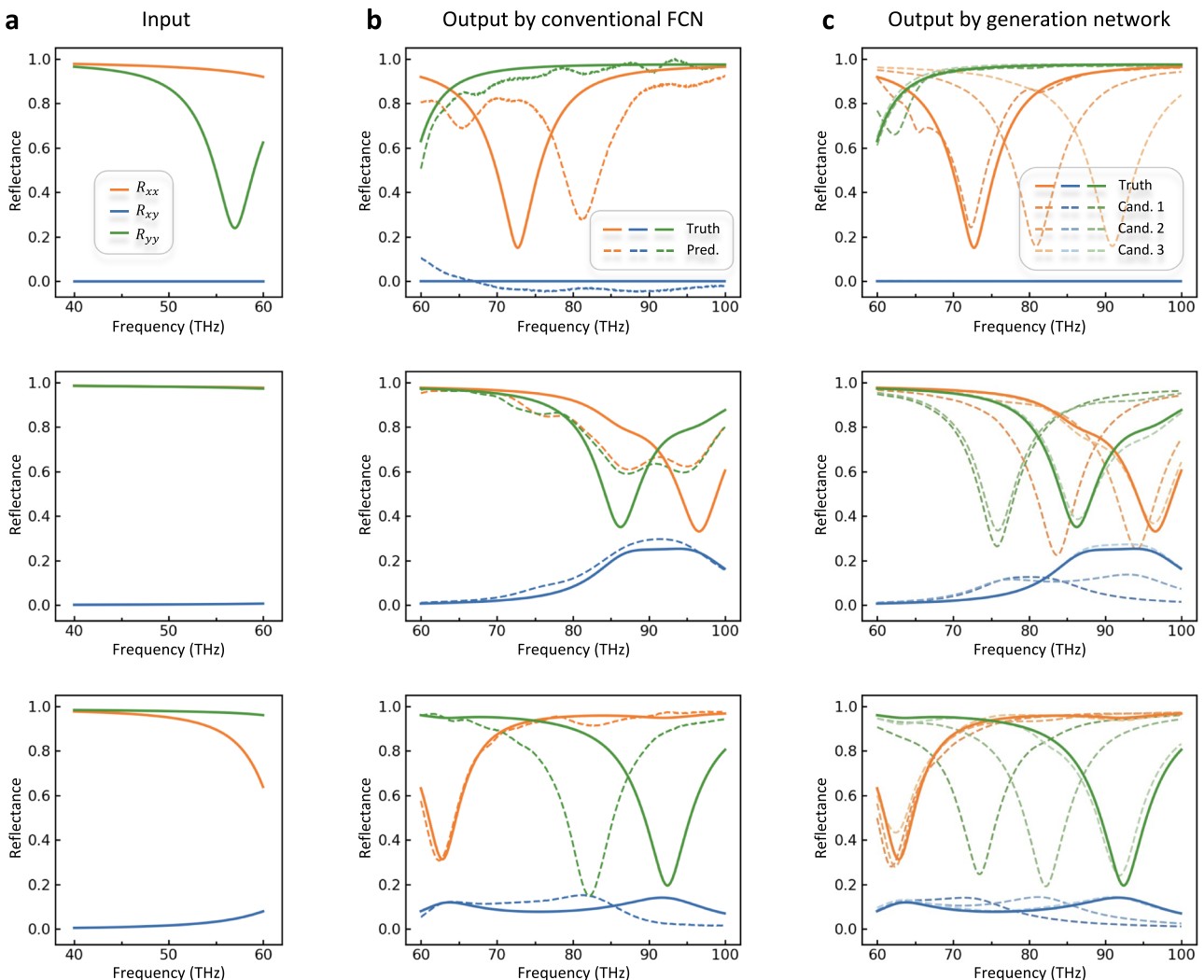

**Fig. 4 | Result comparison with different design strategies. a** The input spectra of three randomly selected samples from the testing dataset, where three reflection coefficients, $R_{xx}$, $R_{xy}$ and $R_{yy}$, are plotted for each sample. **b** The inconsistent results when conventional fully connected network is used to handle the non-convergence spectra-to-spectra problem. The solid and dashed lines represent the ground truth and predicted results from the fully connected network, respectively. **c** The results with our VAE-based generation network. The dashed lines represent different candidates generated by sampling over the latent space, and at least one candidate is matched well with the solid line.

unsupervised model CycleGAN[45], which also minimizes the distance between the original data and the cycle-transferred data in order to preserve content consistency while performing image style transfer. It requires the data from each domain to possess inherent shared characteristics (i.e., style). By contrast, we use "cycle consistency" for the precise matching of data pairs at a microscopic level of complex many-to-many mapping. The value of the minimal distance will also indicate the rationality of the obtained optimal solution for any random input. Figure 5b shows the final solutions of spectra-to-spectra prediction. With the same input in Fig. 4a, the final solution is closely matched with the ground truth in all cases. The satisfactory results imply that the elimination network is competent for picking out the parent node by producing the optimal leaf node.

To quantify the outstanding performance of our framework, we quantitatively define three criteria as follows. (1) MSE: the mean square error between the predicted spectra and ground-truth spectra. (2) Average accuracy $(1 - e_{ave}) \times 100\%$, where $e_{ave}$ is defined as the average relative error between the predicted spectra and ground-truth spectra, that is, $\frac{1}{n}\sum_{i=1}^{n}|y_i - y_i'|/y_i$, where $y_i(y_i')$ represents the $i$th data point of

the ground-truth (predicted) spectra, and $n$ is the number of spectral points. (3) Similarity: the correlation coefficient between two vectors/curves, defined as,

$$\text{Similarity} = \frac{\sum_{i=1}^{n}(y_i - \bar{y}) \times (y_i' - \bar{y}')}{\sqrt{\sum_{i=1}^{n}(y_i - \bar{y})^2} \times \sqrt{\sum_{i=1}^{n}(y_i' - \bar{y}')^2}} \times 100\% \quad (2)$$

where $\bar{y}(\bar{y}')$ represents the mean of the ground-truth (predicted) spectra over all points $y_i(y_i')$. Figure 5c displays the summary statistics of three quantitative criteria on the elliptical dataset when trained with the baseline FCN model and our framework, separately. For the MSE loss, the baseline FCN is nearly one order of magnitude larger than our framework, indicating the failure of convergence. Besides, the average accuracy and the similarity give more intuitive comparisons between the predicted spectra curves and the ground truth. Both criteria exhibit much higher accuracies when our framework is adopted. See Supplementary Note 5 for results on two additional datasets. All of these guarantee the generality and versatility of the proposed framework.

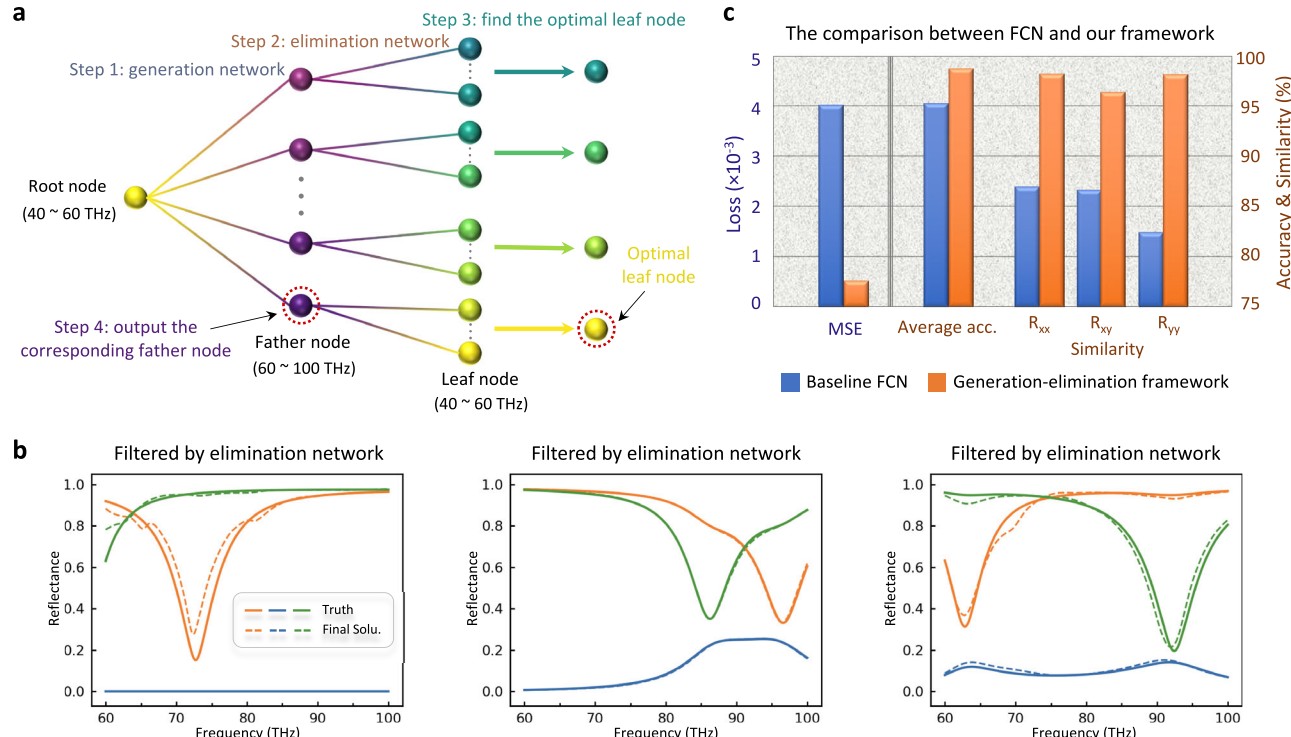

**Fig. 5 | Mimicking the generation-elimination mechanism with a bifurcating tree. a** A tree diagram used to illustrate the generation-elimination mechanism and the birth of the best candidate. The father node whose leaf node has the minimum distance from the root node is chosen as the final output via four steps. **b** The final solutions output by the elimination network, whose inputs are given in Fig. 4a. The solid and dashed lines represent the ground-truth and filtered results by the elimination network, respectively. **c** The quantitative comparisons of three criteria on the elliptical dataset when trained with the baseline FCN model and our framework, separately.

## Discussion

To conclude, we put forward another class of metasurface design and demonstrate its great importance in spectra-to-spectra inference. A novel generation-elimination framework is proposed to cope with the complicated many-to-many mapping dilemma; this is unruly in conventional deep learning algorithms and unrepeatable from widely-studied forward and inverse templates. In our scheme, the stochastically sampled latent variable combined with the input spectra affords the leverage to generate diverse nominations and retain the best one through VAE-based networks, benefiting a lot of applications involving complex physical processes. Furthermore, we come up with an effective approach called auto-encoder to execute dimensionality reduction on continuous and abstract data nonlinearly, which can be adopted in a multitude of data-dependent yet difficult-to-handle visualization scenarios. In addition to characterizing high-frequency spectra, the proposed generation-elimination framework could be readily extended to other research domains of photonic design or even facilitate studies in computer science where conditional variational auto-encoder (CVAE[43]) originates, providing a universal and effective method for fields that are also confronted with bidirectional non-uniqueness predicament. Looking forward, with the nature of physical interpretability elegantly investigated and preserved by the neural networks, we can further facilitate real-world applications, such as illusive cloaking[5,6,46], imaging[47], and wireless communications[38,48,49].

## Methods
### Data collection
In total, 800 elliptical-shaped metasurface patterns are created in Python by sampling over three parameters, i.e., major axis, minor axis, and rotation angle (ranging from 0 to 90 because of its symmetrical property and cross-polarized reflection). To prove the generality of our model, other geometries (cross, h-shape, split-ring, arc, sector and

rectangle, etc.) are also collected by sampling over their parameters; see Supplementary Figs. S6 and S7 and Supplementary Table S5 for intuitional and quantified results of two geometrical patterns. The relevant pattern-generating algorithms and the detailed illustrative procedure are included in Supplementary Note 6. The collected patterns are then transformed into 64 × 64 binary images and matrixes, prepared to be sent into numerical simulations, where "1" stands for gold and "0" stands for air. Instead of using fixed parameterization, we treat metasurfaces as 2D pixel-wise images. In this way, arbitrary design shapes can be properly represented, avoiding the limitations of some existing practices that are only able to represent fixed structures with few geometrical parameters.

### Numerical simulations
In the numerical simulation, we import these binary matrixes into the commercial software package CST Microwave Studio and continuously generate the reflection spectra data using the MATLAB-CST co-simulation method. The metasurface with a period of 2 μm is modeled as a sandwich structure, where the middle spacer is treated as a lossless dielectric with dielectric constant $\varepsilon_r = 4$ and thickness $ts = 100$ nm, and the thickness of the top resonator and the bottom ground plane is fixed as $tm = 50$ nm. The reflection spectra of interest are set in the mid-infrared region from 40 to 100 THz and uniformly discretized into 1001 magnitude data points for each reflection coefficient, constituting a 3003-dimensional vector. The input spectrum, ranging from 40 to 60 THz, is a 999-dimensional vector, while the output spectrum is a 2004-dimensional vector ranging from 60 to 100 THz.

### Training of generation and elimination network
The simulated data are shuffled, and 75% are blindly selected as the training set, and the remaining 25% are used for validation and testing.

Both sub-networks are trained using Python version 3.7.11 and TensorFlow framework version 2.7.0 (Google Inc.) on a local computer (Intel(R) 6-Core CPU i5 @3.7 GHz with 8GB DDR4, running a macOS operating system). It takes nearly an hour for each network to converge after running 10,000 epochs using a batch size of 100 when running on elliptical-shaped metasurfaces. In the future, transfer learning[50–52] could be considered among different geometric pattern groups, and unlabeled spectra data could be utilized in a semi-supervised learning strategy[43].

## Reporting summary

Further information on research design is available in the Nature Portfolio Reporting Summary linked to this article.

## Data availability

Data presented in this publication is available on Figshare with the following identifier (https://doi.org/10.6084/m9.figshare.23601057).

## Code availability

The codes used in the current study are available from the corresponding authors upon request.

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

## Acknowledgements

The work at Zhejiang University was sponsored by the Key Research and Development Program of the Ministry of Science and Technology under Grant Nos. 2022YFA1404704 (H.C.), 2022YFA1405200 (H.C.), and 2022YFA1404902 (H.C.), the National Natural Science Foundation of China (NNSFC) under Grant Nos. 62101485 (C.Q.) and 61975176 (H.C.), the Key Research and Development Program of Zhejiang Province under Grant No. 2022C01036 (H.C.), and the Fundamental Research Funds for the Central Universities (H.C.).

## Author contributions

C.Q., J.C., and H.C. conceived the idea of this research. J.C. performed the network modeling and data analysis. J.Z. and Y.J. provided assistance in machine learning. J.C. and C.Q. wrote the paper. All authors shared their insights and contributed to discussions on the results. C.Q. and H.C. supervised the project.

## Competing interests

The authors declare no competing interests.
