## [Peer Review file · Nature Communications]

REVIEWER COMMENTS

Reviewer #1 (Remarks to the Author):

The authors consider the highly relevant topic of machine learning for (meta-)materials design, where the purpose of ML is to facilitate the search for materials with desired optical responses. The work distinguishes itself from previous works by considering an alternative to common forward and backward designs, and proposes instead to predict the optical response from other correlated optical responses.

The motivations are clearly stated, and the research appears highly relevant to the field. However, the paper has multiple major drawbacks:

1. The complexity of the proposed machine learning method lacks justification. (e.g. Why can't a many-to-many problem be approached by two standard one-to-many networks?) A simple algorithm-independent formulation of the many-to-many problem at hand is missing.
2. There is no systematic quantified evaluation on some representative set of meta-materials or optical responses. Only anecdotal visual evidence is provided for the desired behavior of the proposed ML method. There is insufficient comparisons to simpler ML baselines.
3. There is no clear worked through example of how this method can be concretely applied in practice.

Overall, I believe that manuscript does not meet the quality standards for publication in this journal.

Reviewer #2 (Remarks to the Author):

The authors report a generation-elimination framework and apply it to metasurface design. In particular, a spectra-to-spectra design idea is proposed. To distinguish it from forward and inverse designs, they term it the third-class metasurface design. This idea is innovative and this work is suitable for publication in Nature Communications. I suggest accept with minor revisions. Just some minor comments here.

1. I would like to suggest the authors compare their works with other related literatures to further benchmark the superiority of this work.
2. The establishment of latent space requires further elaboration. Although the latent distribution has been explained in this paper, I think it is still necessary to supplement the rationality of the selection of Gaussian function.
3. The elliptic structure is used as the verification and the supplementary materials provide more structures. Please discuss how different structures affect the generality of the model or explain the applicability of the network in terms of some structures.
4. At the end of the article, the author mentioned the application of transfer learning in the following work. I suggest to add some references on metasurface design via transfer learning.
5. About Supplementary Figs. S5 and S6 for the results of two geometrical patterns (C and H shape), the authors are suggested to explain how to generate a certain type of pattern in this 64×64 binary grid. It is better to add the relevant algorithm or method in this section.

Reviewer #3 (Remarks to the Author):

This paper proposes a neural-network-based framework for addressing the spectra correlation problem and demonstrates some results. The problem the authors are addressing is potentially interesting in various research areas where extrapolation of data to higher or lower frequency ranges is often needed, for example in the application of Kramers-Kronig. The two conditional VAEs function as a model to infer the bidirectional relationship. The mathematical details are not new, but rather follow standard conditional VAE techniques.

The paper cannot be accepted due to the following major issue:

There are no details given of the error between the network prediction and ground truth. The authors only provide visual comparisons. Further, there is no summary statistics given for the test set. Both of these are obligatory in the field of deep learning. Thus, there is no way to judge on the performance of their work.

Other minor points.

Could you provide the details on the architecture of the FCN used for comparison? Could you provide the training loss curves? Does a larger model help with the nonconvergence? If you use three FCNs to represent R_{xx} , R_{xy} and R_{yy} separately, would the ill-conditionness of this correlation map relax?

What is the reconstruction loss on the testing set with your combined two-step networks? In your dataset, the low frequency spectra y has the direct corresponding high spectra x . So, what is the loss when you pass any y to the generator and eliminator to get the optimal output \hat{x} compared with x ?

What is the dimension of the latent space for your other two? I assume latent space with two variables are sufficient is due to your input degree of freedom is only three (two radiuses and a rotation angle). How about larger input dimension, say the periodicity and thickness are changing? Does your latent space visualization method still hold?

It was not clear to me how you constructed the ellipse geometry in CST. In the manuscript, it seems like the geometry gets discretized to finite pixel and import that into CST? Is there a specific reason to discretize those geometries when CST can directly model continuous boundaries?

Response Letter to Reviewers

General comments from Referee #1:

The authors consider the highly relevant topic of machine learning for (meta-)materials design, where the purpose of ML is to facilitate the search for materials with desired optical responses. The work distinguishes itself from previous works by considering an alternative to common forward and backward designs, and proposes instead to predict the optical response from other correlated optical responses.

The motivations are clearly stated, and the research appears highly relevant to the field. However, the paper has multiple major drawbacks:

Authors Response:

We thank the referee for his/her positive recognition of our work. In the following, we will address the specific comments point-by-point whilst revising our manuscript to avoid possible confusion.

Specific comments from Referee #1:

Referee #1 -- Comment 1:

1. The complexity of the proposed machine learning method lacks justification. (e.g. Why can't a many-to-many problem be approached by two standard one-to-many networks?) A simple algorithm-independent formulation of the many-to-many problem at hand is missing.

Authors Responses 1:

We appreciate the referee for the constructive comment. Below we will specifically justify the complexity of our generation-elimination framework, beyond two standard one-to-many networks.

Foremost, we want to illustrate how we realize the many-to-many mapping between correlated optical responses. The trained generation and elimination networks provide two latent spaces that perform one-to-many mappings, and the decoders that can transform the latent variables into their respective spectra spaces (i.e., high-dimensional spectra data). To realize the fusion of two spectra spaces, **a bifurcating tree is exploited to build up a two-way mutual selection between low-frequency and high-frequency spaces**. As illustrated in Fig. R1, for a given input (low-frequency), the generation network can generate various candidates (high-frequency). But, the question is how to determine the optimal candidate. To solve this, the elimination network is used to reversely map each candidate into the original space. Based on the two-step bifurcating tree, the optimal candidate is picked out by calculating the Euclidean distance between the input and secondary

candidates. In this way, the two-way mutual selection and many-to-many mapping are realized without consulting structural information to evaluate the inferred candidates/spectra.

Each standard one-to-many network has an inherent error **because of the deviation of distributions and bias of sampling**. It is worth mentioning the KL divergence loss term in Eq. 1 of the main text, which competes with the reconstruction loss until convergence and forces the true latent distribution to be the same as the standard Gaussian prior distribution. Although it affords the leverage to sample from standard Gaussian distribution during the inference phase and brings diverse nominations, it will inevitably cause errors in the spectral correlation because of the non-negligible deviation of sampling between the true latent distribution and standard Gaussian prior distributions. **Thus, if we directly adopt two one-to-many networks, the error will be accumulated and magnified.** As shown in Fig. R1, the inferior candidates can be excluded by comparing the distances, and the error will be eliminated to the greatest extent through the two-way mutual selection. In traditional inverse design based on one-to-many networks, the screening process that involves structure information is also indispensably needed, otherwise, they cannot evaluate the error and verify the rationality of the solution.

In conclusion, in our framework, we advantageously make full use of the conventional one-to-many network and compensate for its limitation when adapted to (meta-)materials design. The inherent error is delicately eliminated by the mutual selection of two spectra spaces through a bifurcating tree. The idea is not duplicated from but in-turn facilitates the studies in computer science and the framework could be readily extended to other research domains of photonic design.

In the new submission, we have added the above discussion,

On lines 63-66, “Besides, the inherent bias in conventional one-to-many network cannot be evaluated and mitigated in virtue of the structural information anymore. The direct connection of two standard one-to-many networks will cause the magnification of the error.”

On lines 73-75, “the operation process is understandable as a hierarchical bifurcating tree, which builds up a two-way mutual selection between two spaces and eliminates the inherent error to the greatest extent.”

On lines 220-222, “Second, the KL divergence term in Eq. 1 reveals that there is a deviation between the true latent distribution and standard Gaussian prior distribution. It means that the latent variables, though sampled from the acceptable region, are not guaranteed to retrieve highly precise nominations.”

On lines 234-237, “With the delicately designed bifurcating tree, the bidirectional mapping and two-way selection are built up between low-frequency and high-frequency spaces, and the inferior candidates can be excluded by comparing the Euclidean distance between the input and secondary candidates in the low-frequency space.”

Fig. R1 | The two-way mutual selection between low-frequency and high-frequency spaces. The trained generation and elimination networks offer two latent spaces that perform unidirectional one-to-many mappings between two spectra spaces. Based on the two-step bifurcating tree, the two-way mutual selection is realized and the deviation is mitigated to the greatest extent. Dist. is the abbreviation of distance and cand. is the abbreviation of candidata.

Referee #1 -- Comment 2:

2. There is no systematic quantified evaluation on some representative set of meta-materials or optical responses. Only anecdotal visual evidence is provided for the desired behavior of the proposed ML method. There is insufficient comparisons to simpler ML baselines.

Authors Response 2:

Thanks for the good suggestion. As a complement to the visual evidence, we use three criteria to systematically quantify the performance of our framework. Meanwhile, we take the fully-connected network as a simpler ML baseline model and evaluate it with the same criteria to highlight the superiority of our framework.

- (1) MSE: the mean square error between the predicted spectra and ground truth spectra.
- (2) Average accuracy $(1 - e_{ave}) \times 100\%$, where e_{ave} is defined as the average relative error between the predicted spectra and ground truth spectra, that is, $\frac{1}{n} \sum_{i=1}^n |y_i - y'_i| / y_i$, where $y_i(y'_i)$ represents the i th data point of the ground truth (predicted) spectra, and n is the number of spectral points.
- (3) Similarity: the correlation coefficient between two vectors/curves, defined as,

$$\text{Similarity} = \frac{\sum_{i=1}^n (y_i - \bar{y}) \times (y'_i - \bar{y}')}{\sqrt{\sum_{i=1}^n (y_i - \bar{y})^2} \times \sqrt{\sum_{i=1}^n (y'_i - \bar{y}')^2}} \times 100\% \quad (\text{R1})$$

where $\bar{y}(\bar{y}')$ represents the mean of the ground truth (predicted) spectra over all points $y_i(y'_i)$.

Table R1 displays the summary statistics of three quantitative criteria on the elliptical dataset when trained with the baseline FCN model and our framework, separately. For the MSE loss, the baseline FCN is nearly one order of magnitude larger than our framework, indicating the failure of convergence and the invalidity of tackling the non-uniqueness predicament. Besides, the average accuracy and the similarity give more intuitive comparisons between the predicted spectra curves and the ground truth. Both criteria exhibit much higher accuracies when our framework is adopted.

Furthermore, we have also shown the quantified summary statistics on another two geometry patterns (arc and distorted h-shape); see **Referee #2 -- Comment 3** in Table R2 on page R8. Both the low MSE errors and high accuracies in all geometry cases further exemplify the generality and versatility of our framework.

In the new submission, we have incorporated Table R1 as Fig. 5c and added the above description,

On lines 19-20, “where the inaccessible spectra are precisely forecasted without consulting structural information, reaching an accuracy of 98.77%.”

On lines 241-255, “To quantify the outstanding performance of our framework, ... generality and versatility of the proposed framework.”

On lines 469-470, “The quantitative comparisons of three criteria on the elliptical dataset when trained with the baseline FCN model and our framework, separately.”

Table R1 | The quantitative comparisons between baseline FCN and our framework.

	MSE	Average acc.	Similarity		
			R_{xx}	R_{xy}	R_{yy}
Baseline FCN	4.015e-3	95.25%	86.96%	86.68%	82.42%
Our framework	5.174e-4 ↓	98.77% ↑	98.13% ↑	96.38% ↑	98.15% ↑

Referee #1 -- Comment 3:

3. There is no clear worked through example of how this method can be concretely applied in practice.

Authors Response 3:

Thanks for the good question. In this work, we proposed a generation-elimination framework for the spectral correlation, where the high-frequency response of the metasurfaces is inferred from the low-frequency response with an accuracy of 98.77%. The major advantage is to address many-to-many mapping in deep learning. **Such advantage brings many benefits.** First, largely reducing the meshed design space and simulation time by only utilizing low-frequency data. Second, extracting the desired images, spectra, and material features from other easily accessible information at a low cost [*Appl. Sci. Res.* **10**, 5790–5795 (2020)].

Apart from its application in metasurfaces, the spectral correlation can be generalized in various fields. To be specific, we use synthetic aperture radar (SAR) to show how this method can be concretely applied in practice, as shown in Fig. R2. Typically, the SAR system generates copious amounts of complex raw data that require intense computation to produce high-resolution images. In some real-time applications, such as navigation decisions, interplanetary missions, and unmanned aerial vehicles, the resolution must be sacrificed for on-the-fly imaging processing, resulting in fuzzy images like the low-resolution rabbit picture in Fig. R2. However, by using our spectral correlation technique, the low-resolution rabbit picture obtained from the SAR system is possible to be transitioned to the high-resolution one with negligible processing time [*Proc. SPIE* **8714**, 871416 (2013)]. For more applications, such as in communications, the noise in the high-frequency band could be wiped off in conjunction with the low-frequency correlation [*JCASSP* **636–640** (2018)]. Also, as pointed out by

Referee #3, “[the problem] is potentially interesting in various research areas where extrapolation of data to higher or lower frequency ranges is often needed, for example in the application of Kramers-Kronig”.

In the new submission, we have added the above discussion,

On lines 55-57, “This is instrumental in a wide range of applications, for example, to extract the desired images, spectra, and material features from other easily accessible information with the advantages of low cost and easy measurement²⁵.”

On lines 88-90, “For example, in communication and Raman spectroscopy, in conjunction with the low-frequency correlation, the noise in the high-frequency region can be largely wiped off³⁹⁻⁴⁰.”

On lines 92-94, “And as a more concrete example, the low-resolution images obtained from synthetic aperture radar (SAR) system could be transitioned to the high-definition one with negligible processing time for some real-time applications⁴².”

Fig. R2 | The application of spectral correlation in the SAR system. With the spectral correlation technique, the low-resolution rabbit image obtained by low-frequency SAR detection is possible to be converted into a high-resolution image.

Referee #1 -- Comment 4:

Overall, I believe that manuscript does not meet the quality standards for publication in this journal.

Authors Response 4:

We apologize for the possible confusion on the practicality and technical issues, and sincerely appreciate the referee for all the constructive comments as they help strengthen the idea presented in the paper. In this work, we for the first time dabble in this many-to-many mapping dilemma and propose a groundbreaking and effective method to cope with it. From the methodology, the inherent error in conventional one-to-many network is delicately eliminated by the mutual selection of two spectra spaces. From the application viewpoint, we conceptualize the third-class metasurface design where the correlated metasurface spectra are precisely forecasted, and we believe it will bring widespread and far-reaching implications to various research domains where extrapolation of spectra data is needed or intractable many-to-many correspondences are involved. We anticipate the response and the updated manuscript could remove the referee’s confusion.

General comments from Referee #2:

The authors report a generation-elimination framework and apply it to metasurface design. In particular, a spectra-to-spectra design idea is proposed. To distinguish it from forward and inverse designs, they term it the third-class metasurface design. This idea is innovative and this work is suitable for publication in Nature Communications. I suggest accept with minor revisions. Just some minor comments here.

Authors Response:

We thank the referee for his/her positive comments and for identifying the novelties of our work. In the following, we will address the specific questions point-by-point.

Specific comments from Referee #2:

Referee #2 -- Comment 1:

1. I would like to suggest the authors compare their works with other related literatures to further benchmark the superiority of this work.

Authors Response 1:

Thanks for the good suggestion. We have followed the referee's suggestion and compared our work with other related literatures [e.g., *Adv. Sci.* **6**, 1900128 (2019), *Nanophotonics* **9**, 1041–1057 (2020), *J. Phys. D-Appl. Phys.* **53**, (2020) & *Adv. Funct. Mater.* **31**, 2101748 (2021)] to highlight the superiority of our work on pages 2-3 in the new main text; see below.

On lines 58-63, "Achieving third-class metasurface design faces a formidable challenge because it involves complex many-to-many mapping, rather than simply one-to-one and one-to-many mapping^{26,27} ... In doing so, conventional deep learning algorithms will become conflicted on how to adjust learnable weights and the convergence can not be guaranteed^{28,29}."

Referee #2 -- Comment 2:

2. The establishment of latent space requires further elaboration. Although the latent distribution has been explained in this paper, I think it is still necessary to supplement the rationality of the selection of Gaussian function.

Authors Response 2:

We appreciate the referee for pointing this out. Actually, the normal distribution is not the only option that can be employed for latent variables in VAEs. There are many other options, such as von Mises-Fisher distribution (hyperspherical VAEs) [arXiv:1804.00891], and Gaussian mixtures [arXiv:1611.02648]. As mentioned in the pioneering VAE work [arXiv:1312.6114], "**under certain mild conditions** outlined in section 2.4 for a chosen approximate posterior $q_\varphi(z|x)$, we can reparameterize the random variable $\tilde{z} \sim q_\varphi(z|x)$ using a differentiable transformation $g_\varphi(\epsilon, x)$ of an (auxiliary) noise variable ϵ " and one of the conditions says "for any 'location-scale' family of distributions we can choose the standard distribution (with location = 0, scale = 1) as the auxiliary variable ϵ , and let $g(\cdot) = \text{location} + \text{scale} \cdot \epsilon$. **Examples: Laplace,**

Elliptical, Student's t, Logistic, Uniform, Triangular and Gaussian distributions". This VAE paper lists many other distributions as long as the conditions are met for the reparameterization trick.

Let's go back to our work. From the theoretical viewpoint, it does not matter so much what distribution latent variables follow, because the non-linear decoder can mimic arbitrarily complicated distribution of observations. However, the normal distribution is still the most widely used in subsequent studies because it has many nice properties, such as analytical evaluation of the KL divergence in the variational loss (Eq. S9-S11 in the supplementary materials), and efficient gradient computation of the reparameterization trick. Moreover, one of the apparent advantages of VAE is that it allows the generation of new samples by sampling in the latent space—which would be quite easy if it follows Gaussian distribution.

In the new submission, we have added the above discussion on lines 134-136,

"We choose Gaussian as the prior distribution because of its analytical evaluation of the variation loss (see Eq. S9-S11 in Supplementary Note 1), and the convenience of performing sampling in the latent space."

Referee #2 -- Comment 3:

3. The elliptic structure is used as the verification and the supplementary materials provide more structures. Please discuss how different structures affect the generality of the model or explain the applicability of the network in terms of some structures.

Authors Response 3:

Thanks for the suggestion. To verify the generality of our model, we use three quantitative indicators to evaluate the performance of our model on different structures. (1) MSE, the mean square error between the predicted spectra and ground truth spectra. (2) Average accuracy, $(1 - e_{ave}) \times 100\%$, where e_{ave} is defined as the average relative error between the predicted spectra and ground truth spectra, that is, $\frac{1}{n} \sum_{i=1}^n |y_i - y'_i| / y_i$, where $y_i(y'_i)$ represents the i th data point of the ground truth (predicted) spectra, and n is the number of spectral points. (3) Similarity, the correlation coefficient between two vectors/curves, defined as,

$$\text{Similarity} = \frac{\sum_{i=1}^n (y_i - \bar{y}) \times (y'_i - \bar{y}')}{\sqrt{\sum_{i=1}^n (y_i - \bar{y})^2} \times \sqrt{\sum_{i=1}^n (y'_i - \bar{y}')^2}} \times 100\% \quad (\text{R2})$$

where $\bar{y}(\bar{y}')$ represents the mean of the ground truth (predicted) spectra over all points $y_i(y'_i)$.

Table R2 displays the results on three different structures. Whatever the geometry type is, the MSE errors are quite low, the average accuracy and the similarity are relatively high. All of these guarantee the generality and versatility of the proposed network and exemplify that the type of structure does not affect the accuracy of our model.

In the new submission, we have incorporated Table R2 as Table S5 and the above discussion in Supplementary Note 5, and noted them in the main text,

On lines 254-255, "See Supplementary Note 5 for results on two additional datasets. All of these guarantee the generality and versatility of the proposed framework."

On lines 278-279, “see Supplementary Figs. S6 and S7 and Table R5 for intuitional and quantified results of two geometrical patterns.”

Table R2 | The quantitative evaluation of the framework on various structures.

Structure	MSE	Average acc.	Similarity		
			R _{xx}	R _{xy}	R _{yy}
Ellipse	5.174e-4	98.77%	98.13%	96.38%	98.15%
Arc	1.066e-3	97.85%	95.24%	96.84%	96.20%
Distorted h-shape	3.516e-4	98.57%	96.14%	98.56%	95.84%

Referee #2 -- Comment 4:

4. At the end of the article, the author mentioned the application of transfer learning in the following work. I suggest to add some references on metasurface design via transfer learning.

Authors Response 4:

We appreciate the referee for pointing this out. As suggested, we have added the literatures [*Nat. Commun.* **12**, 2974 (2021) and *Phys. Rev. Appl.* **18**, 024022 (2022)] closely related to the metasurface design via transfer learning in the new submission, and noted them as,

On lines 301-303, “In the future, transfer learning⁴⁷⁻⁴⁹ could be considered among different geometric pattern groups and ...”

Referee #2 -- Comment 5:

5. About Supplementary Figs. S5 and S6 for the results of two geometrical patterns (C and H shape), the authors are suggested to explain how to generate a certain type of pattern in this 64 × 64 binary grid. It is better to add the relevant algorithm or method in this section.

Authors Response 5:

We thank the referee for the suggestion and have appended the relevant algorithms in the corresponding section. For the elliptical patterns, we first sample over its three parameters (i.e., major axis, minor axis, and rotation angle) with *for* loops, and then invoke the *cv2.ellipse* function to draw the geometry. Similarly, for the arc and distorted h-shaped patterns, we sample over their corresponding parameters in *for* loops and call *matplotlib.pyplot.Arc* and *cv2.rectangle* functions to draw the geometries. See below for three relevant and complete algorithms that are written in pseudo-code; the main built-in packages (including rotation and distortion) are underlined for clarity.

In the new submission, we have appended these pattern-generating algorithms in Supplementary Note 6, and noted them in the main text.

On lines 279-280. “The relevant pattern-generating algorithms and the detailed illustrative procedure are included in Supplementary Note 6.”

Algorithm 1 Ellipse Pattern Generation

```
1: for axesLength_X = 2, 4, ..., 28, 30 do
2:   for axesLength_Y = 2, 4, ..., 28, 30 do
3:     for angle = 0, 10, ..., 80 do
4:       Initialize img  $\leftarrow$  64  $\times$  64 zeros
5:       img = cv2.ellipse(img, axesLength_X, axesLength_Y, angle)
6:       Export and save img as 64  $\times$  64 binary matrix
7:       if axesLength_X == axesLength_Y (degrade as circle) then
8:         Break
9:       end if
10:    end for
11:  end for
12: end for
```

Algorithm 2 Arc Pattern Generation

```
1: for diameter, start_angle, total_angle, linewidth do
   $\triangleright$  all parameters are adequately and randomly sampled within a reasonable range
2:   width, height  $\leftarrow$  diameter
   $\triangleright$  here we only exhibit the main parameters when calling functions
3:   img = matplotlib.pyplot.Arc(center, width, height, start_angle, theta1 = 0, theta2 = total_angle, linewidth)
4:   img = cv2.resize(img, dsiz = (64, 64))
5:   Transform img into binary image
6:   Export and save img as 64  $\times$  64 binary matrix
   $\triangleright$  for Sector Pattern Generation, invoke matplotlib.pyplot.Wedge in the same way
7: end for
```

Algorithm 3 Distorted H-shape Pattern Generation

```
1: function H_SHAPE_DRAWING(main_comb, side_comb1, side_comb2, angle, add_angle)
2:   Initialize img  $\leftarrow$  64  $\times$  64 zeros
3:   for comb = main_comb, side_comb1, side_comb2 do
4:     vertexes  $\leftarrow$  comb
5:     img = cv2.rectangle(img, vertexes)
6:   end for
   $\triangleright$  apply add_angle for angle deviation between two side_combs
   $\triangleright$  apply angle for rotation of the overall shape
   $\triangleright$  here we only exhibit the main parameters when calling functions
7:   RotationMatrix = cv2.getRotationMatrix2D(center, angle)
8:   img = cv2.warpAffine(img, RotationMatrix)
  return img
9: end function

10: for main_width, main_height, side_width1, side_width2, side_height, angle, add_angle, distort_degree do
   $\triangleright$  all parameters are adequately and randomly sampled within a reasonable range
11:   main_comb = (main_width, main_height)
12:   side_comb1 = (side_width1, side_height)
13:   side_comb2 = (side_width2, side_height)
14:   img = H_SHAPE_DRAWING(main_comb, side_comb1, side_comb2, angle, add_angle)
15:   Using the Wand package to distort the img
16:   img.distort(method = 'arc', (distort_degree,))
17:   Reshape img back after distortion
18:   Export and save img as 64  $\times$  64 binary matrix
19: end for
```

General comments from Referee #3:

This paper proposes a neural-network-based framework for addressing the spectra correlation problem and demonstrates some results. The problem the authors are addressing is potentially interesting in various research areas where extrapolation of data to higher or lower frequency ranges is often needed, for example in the application of Kramers-Kronig. The two conditional VAEs function as a model to infer the bidirectional relationship. The mathematical details are not new, but rather follow standard conditional VAE techniques.

Authors Response:

We thank the referee for the encouraging comments and for pointing out some novelties of our work. In the following, we will address the specific comments point-by-point whilst revising our manuscript.

Specific comments from Referee #3:

Referee #3 -- Comment 1:

The paper cannot be accepted due to the following major issue:

There are no details given of the error between the network prediction and ground truth. The authors only provide visual comparisons. Further, there is no summary statistics given for the test set. Both of these are obligatory in the field of deep learning. Thus, there is no way to judge on the performance of their work.

Authors Response 1:

Thanks for the good question. As a indispensable complement to the visual comparisons, we define three indicators to systematically quantify the performance of our framework. (1) MSE: the mean square error between the predicted spectra and ground truth spectra. (2) Average accuracy $(1 - e_{ave}) \times 100\%$, where e_{ave} is defined as the average relative error between the predicted spectra and ground truth spectra, that is, $\frac{1}{n} \sum_{i=1}^n |y_i - y'_i| / y_i$, where $y_i(y'_i)$ represents the i th data point of the ground truth (predicted) spectra, and n is the number of spectral points. (3) Similarity: the correlation coefficient between two vectors/curves, defined as,

$$\text{Similarity} = \frac{\sum_{i=1}^n (y_i - \bar{y}) \times (y'_i - \bar{y}')}{\sqrt{\sum_{i=1}^n (y_i - \bar{y})^2} \times \sqrt{\sum_{i=1}^n (y'_i - \bar{y}')^2}} \times 100\% \quad (\text{R3})$$

where $\bar{y} (\bar{y}')$ represents the mean of the ground truth (predicted) spectra over all points $y_i(y'_i)$.

Table R3 displays the summary statistics of these three indicators for the test set. From the table, the low MSE error, the high average accuracy, and the similarity quantitatively provide solid evidence about the outstanding performance of our framework.

In the new submission, we have added Table R3 as Fig. 5c and above discussion,

On lines 19-20, “where the inaccessible spectra are precisely forecasted without consulting structural information, reaching an accuracy of 98.77%.”

On lines 241-255, “To quantify the outstanding performance of our framework, ... generality and versatility of

the proposed framework.”

On lines 469-470, “The quantitative comparisons of three criteria on the elliptical dataset when trained with the baseline FCN model and our framework, separately.”

Table R3 | Three indicators to quantitatively evaluate the performance of our framework.

MSE	Average acc.	Similarity		
		R_{xx}	R_{xy}	R_{yy}
5.174e-4	98.77%	98.13%	96.38%	98.15%

Referee #3 -- Comment 2:

Other minor points.

Could you provide the details on the architecture of the FCN used for comparison? Could you provide the training loss curves? Does a larger model help with the nonconvergence? If you use three FCNs to represent R_{xx} , R_{xy} and R_{yy} separately, would the ill-conditionness of this correlation map relax?

Authors Response 2:

We appreciate the referee for raising these questions. Actually, we have tried FCN at the very beginning, but found **it fails to solve our question** due to the existing many-to-many issue in spectral correlation. Below, we will specifically address these raised questions one-by-one.

The detailed architecture of FCN is shown in Table R4. The baseline model is composed of 9 fully connected layers with 300 hidden neurons in each layer. Correspondingly, the training (orange line) and validation (blue line) loss curves are plotted in Fig. R3. The relatively large fluctuations in the loss curves indicate the failure of convergence. In other words, even though the loss curves incline, we cannot read from which point (the convergence point) the error is within a relatively small range. Besides, the validation error stays at a high level (around $4e^{-3}$ measured in MSE), compared to $5e^{-4}$ in our proposed framework. The error of nearly one order of magnitude not only proves that the FCN fails to converge, but also exposes its performance gap with our framework when confronted with the same bidirectional non-uniqueness predicament.

As suggested, we further increase the number of layers to 10 and 11 as larger models and train them for 30,000 epochs. The complete learning loss curves are plotted in Fig. R4. The non-convergence problem is not eased with the validation error still vigorously fluctuating at a relatively high value (around $5e^{-3}$ measured in MSE).

For the last question “*If you use three FCNs ...*”, we use three independent FCN networks (each with the same configuration as the baseline model in Table R4, except for the input size and output size) to train R_{xx} , R_{xy} and R_{yy} , separately. Table R5 displays the numerical results of utilizing the three indicators to evaluate the performance of baseline FCN, three FCNs and our framework. The close and relatively low values of baseline FCN and three FCNs indicate that the strategy of using three networks to represent R_{xx} , R_{xy} , R_{yy} , separately, does not alleviate the spectral correlation mapping issue. Besides, the quantitative differences between three FCNs and our framework indicate that the performance gap still exists.

Overall, we believe and have proven that the FCN model is bound to be in the cart when faced with complicated many-to-many mapping. And from the theoretical viewpoint, it is incompetent of tackling the non-convergence problem.

In the new submission, we have added Table R4 as Table S3, Fig. R3 as Fig. S3, Table R5 as Table S4, and the above discussion in Supplementary Note 2, and noted them in the main text,

On lines 207-208, “see Supplementary Note 2 for more details.”

Table R4 | Architecture and parameters of the baseline model FCN used for comparison.

layer	name	op.	size-in	size-out
1	Input	-	2004	-
2	dense1	fc relu	2004	300
3	dense2	fc relu	300	300
4	dense3	fc relu	300	300
5	dense4	fc relu	300	300
6	dense5	fc relu	300	300
7	dense6	fc relu	300	300
8	dense7	fc relu	300	300
9	dense8	fc relu	300	300
10	prediction	fc linear	300	999

Fig. R3 | The training and validation loss of FCN measured in MSE. The blue line is the validation loss and the orange line is the training loss. Both curves have been smoothed with a smoothing factor = 0.4. The validation error fluctuates vigorously above $4e^{-3}$.

Fig. R4 | The training and validation loss with larger models of FCN measured in MSE. The upper and bottom charts are the loss curves of larger baseline models with 10 and 11 layers, respectively. For each, the blue line is the validation loss and the orange line is the training loss. Both curves have been smoothed with a smoothing factor = 0.4.

Table R5 | The quantitative comparisons between baseline FCN, three FCNs and our framework.

	MSE	Average acc.	Similarity		
			R_{xx}	R_{xy}	R_{yy}
Our framework	5.174e-4	98.77%	98.13%	96.38%	98.15%
Baseline FCN	4.015e-3	95.25%	86.96%	86.68%	82.42%
Three FCNs	3.858e-3	93.50%	84.37%	86.21%	84.94%

Referee #3 -- Comment 3:

What is the reconstruction loss on the testing set with your combined two-step networks? In your dataset, the low frequency spectra y has the direct corresponding high spectra x . So, what is the loss when you pass any y to the generator and eliminator to get the optimal output \hat{x} compared with x ?

Authors Response 3:

This is a very good question. Firstly, we want to explain the training process and the reconstruction loss of our

generation-elimination framework. As shown in Fig. R5a, such framework consists of two cascaded networks, namely, generation network and elimination network. We train these two networks independently with their own reconstruction loss, or more accurately, with their own objective loss function. For the generation network, it takes both x (high spectra) and y (low spectra, the label) as inputs, and finally decoded into x' . The reconstruction loss of the generation network is calculated here as the distance between x and x' , which is only used to update the generation network. Similarly, for the elimination network, it takes both x (the label) and y as inputs, and finally decoded into y' . The elimination network's reconstruction loss is calculated as the distance between y and y' , and used to update the elimination network.

When combining these two networks together, we do not need to consider additional reconstruction loss. The two networks are combined by a hierarchical bifurcating tree without further joint training. After both sub-networks have been trained independently, only their decoders are retained and will be utilized in the inference phase, as implied in Fig. R5a. For any input y_{test} , it will firstly be decoded into father nodes (x'_{test}) by the generator's decoder, and then each father node will give birth to leaf nodes (y'_{test}) by the eliminator's decoder. Referring to the tree diagram in Fig. R5b, the optimal output x'_{test} is selected by finding the minimum distance between the root node (input y_{test}) and leaf nodes (y'_{test}). Besides, the value of the minimal distance would indicate the compatibility of the obtained optimal solution, that is, whether there should be a corresponding reasonable and logical output for the random input.

In the new submission, we have clarified this point,

On lines 168-170, "When two cascaded networks have been trained independently, they will be combined by a hierarchical bifurcating tree without further joint training."

On lines 233-234, "Besides, the value of the minimal distance will indicate the rationality of the obtained optimal solution for any random input."

Fig. R5 | Illustrative procedure of the training and inference phases. a, Only two decoders are preserved after

the independent training of two sub-networks. The two decoders are combined with a hierarchical bifurcating tree without further training. **b**, The bifurcating tree that combines the generator's and eliminator's decoders. The optimal solution x'_{test} is selected by calculating and finding the minimal distance between the root node y_{test} and leaf nodes y'_{test} .

Referee #3 -- Comment 4:

What is the dimension of the latent space for your other two? I assume latent space with two variables are sufficient is due to your input degree of freedom is only three (two radiuses and a rotation angle). How about larger input dimension, say the periodicity and thickness are changing? Does your latent space visualization method still hold?

Authors Response 4:

Thanks for the good question. We understand that “your other two” pointed out by the referee refers to another two geometries, the arc and distorted h-shape geometrical patterns, which are also trained with two dimensions (the same setting as the elliptical one). Here, we set the dimension as two merely because the latent space consisting of two-dimensional standard Gaussian variables can achieve relatively high accuracy, and the input degree of freedom (DOF) is not the main factor that determines the dimension. We trained our model with the dimensionality of 5 and 10, separately, on the elliptical dataset. The results can be seen in Table R6. For both generative and eliminative networks, the ultimate validation errors are very similar in all cases, which verifies two dimensions are sufficient for the elliptical dataset. Considering the time cost and efficiency maximization, we set the dimension of our latent space as two.

As for the larger input dimension, taking the distorted h-shape as an example, its DOF is at least eight (main_width, main_height, side_width1, side_width2, side_height, angle, add_angle, distort_degree). We also trained it with two dimensions and achieved an accuracy of 98.57%.

Finally, we want to note that, in our latent space visualization task, the difficulty does not lie in the dimension of the latent space, but in how to leverage the abstract high-dimensional spectra data to colorize/feature the latent space, to further prove the sub-network can conditionally encode the input and output spectra into a compact but informative space. Furthermore, when the dimension of latent space is larger than two, we can still use a dimensionality reduction approach, such as PCA or t-SNE method (and also the auto-encoder approach we use to extract spectra representation), to reduce the dimensionality to two or three for visualization. In the inverse design work [*Adv. Mater.* **31**, 1901111 (2019)], the latent space dimension is reduced from 20 to 2 by using the t-SNE method for visualization.

In the new submission, we have clarified this point in Supplementary Note 5,

On page 11, “Even though the degree of freedom is increased (at least eight for the distorted h-shape pattern) for both cases, the dimension of the latent space is still set as two and can achieve relatively high accuracy.”

And added it the main text,

On lines 138-139, “considering the time cost and efficiency maximization”

Table R6 | The ultimate validation errors of two sub-networks upon different dimensionality values

	2-dim	5-dim	10-dim
Generative network	566.4963	566.0163	564.6725
Elimination network	352.4160	352.4380	352.1880

Referee #3 -- Comment 5:

It was not clear to me how you constructed the ellipse geometry in CST. In the manuscript, it seems like the geometry gets discretized to finite pixel and import that into CST? Is there a specific reason to discretize those geometries when CST can directly model continuous boundaries?

Authors Response 5:

Thanks for the careful reading. Regarding the question “*It was not clear to me...import that into CST?*”, the answer is yes. The detailed procedure of constructing the ellipse geometry (as well as other geometries) is depicted in Fig. R6; see below.

Step 1: We firstly create the binary black/white images in Python by sampling over all possible design parameters, such as length, width, diameter and rotation angle of different parts of the geometries, utilizing built-in Python packages such as *Matplotlib* and *OpenCV* to draw these binary images. The detailed algorithms with the pseudo-code are elaborated in **Referee #2 -- Comment 5** on pages R8-R9.

Step 2: Then, these binary images are exported to individual .txt files as 64×64 matrixes (i.e., one matrix in a file), which are prepared to be taken as inputs in the next step using the MATLAB-CST co-simulation method.

Step 3: In the numerical simulation, MATLAB reads these binary matrixes from.txt files and constructs them in CST, where ‘1’ stands for gold and ‘0’ stands for air. Other parameters such as periodicity and thickness are also pre-defined in MATLAB (see Methods in the main text for more detail), to control the modeling in CST and then numerically calculate the spectra.

Fig. R6 | Illustrative procedure of constructing the ellipse geometry. The three steps to construct elliptical-shaped metasurface patterns in CST and obtain the reflection spectra (the same procedure for the other geometries). The first two steps are executed in Python and the third step is performed by MATLAB-CST co-simulation.

In the new submission, we have incorporated Fig. R6 as Fig. S8 and above procedure in Supplementary Note 6, and noted them in the main text,

On lines 279-280, “The relevant pattern-generating algorithms and the detailed illustrative procedure are included in Supplementary Note 6.”

Regarding the question “Is there a ...”, we will explain it in the following:

When CST models the geometries, they are actually constructed as discrete pixels instead of continuous boundaries, yet have a higher resolution compared with our 64×64 binary images. Fig. R7 shows the simulated spectra of the same pattern constructed in two different methods. The negligible error between the two spectra curves verifies the feasibility of our discretization method. Besides, the discretization method is actually widely used in intelligent metasurface designs [e.g. *Adv. Mater.* **31**, 1901111 (2019), *Nano Lett.* **18**, 6570–6576 (2018)]. It allows for the generation of essentially arbitrary geometries. In our scenario, taking the distorted h-shape pattern as an example, we invoke the built-in packages in Python to control the degree of distortion, which can hardly be described by a few fixed parameters. That is, the discretization method not only facilitates the data collection, but also helps enrich and diversify the dataset and thus validate the generalization of our model.

In the new submission, we have added the above discussion on lines 282-285,

“Instead of using fixed parameterization, we treat metasurfaces as 2D pixel-wise images. In this way, arbitrary design shapes can be properly represented, avoiding the limitations of some existing practices that are only able to represent fixed structures with few geometrical parameters.”

Fig. R7 | The simulation results of two patterns generated by the parameterization and discretization methods, separately.

REVIEWER COMMENTS

Reviewer #1 (Remarks to the Author):

The authors addressed my main concerns, in particular, clarifying the purpose and meaning of the many-to-many approach, and providing quantified evidence that the many-to-many approach leads to a significant accuracy improvement. However, I think the paper could still be much improved w.r.t. clarity, structure, and discussion of related work. Furthermore, results would benefit from being further extended.

Detailed comments:

- The proposed ML approach seem to be related to the CycleGAN, in particular, to the concept of “cycle consistency”. It seems to me that the bifurcation tree selection can be seen as a cycle consistency criterion applied at test time. It would be good to mention this related work in order to highlight that there are further ML tools that could potentially be investigated for such many-to-many mappings.
- The results section currently discusses the VAE/ELBO extensively. However, since the VAE/ELBO is an existing method, I think that such discussion would fit better in the methods section at the end of the paper. I would expect the results section to be more focused on the way the two VAEs are used together for the purpose of the application, in particular, highlighting the interconnection between the two VAEs and the bifurcation tree (cycle consistency) criterion.
- Furthermore, the clarity of the technical section could be improved. Currently, figures do not include the notation and terms used in the paper, and therefore, they do not facilitate the understanding of the technical sections of the paper. Some of the figures are unintuitive, for example, it took me time to realize in Fig. 2a (top) that the blue box is what we actually feed as input, and that the orange boxes at the input and output is some kind of recurrent connection. Maybe a more abstract subfigure showing in a very simple way how the two VAEs interconnect would be useful. Actually, I like Fig. R1 from the authors’ response letter, and think it could potentially be used for the main paper.
- While the authors have added a quantified comparison, I think the results section remains a bit thin, in particular, it would be good to have at least two distinct data generation / numerical simulation scenarios in order to demonstrate that the high reported performance is reproducible under slightly varying conditions.

Reviewer #2 (Remarks to the Author):

All I concerned in my last comments have been addressed in this revised manuscript, which is quite satisfying. I recommend acceptance of the revised version.

Response Letter to Reviewers

General comments from Referee #1:

The authors addressed my main concerns, in particular, clarifying the purpose and meaning of the many-to-many approach, and providing quantified evidence that the many-to-many approach leads to a significant accuracy improvement. However, I think the paper could still be much improved w.r.t. clarity, structure, and discussion of related work, Furthermore, results would benefit from being further extended.

Authors Response:

We appreciate the referee for his/her careful reading and acknowledgement of our efforts. In the following, we address the specific comments point-by-point whilst revising our manuscript.

Specific comments from Referee #1:

Referee #1 -- Comment 1:

The proposed ML approach seem to be related to the CycleGAN, in particular, to the concept of "cycle consistency". It seems to me that the bifurcation tree selection can be seen as a cycle consistency criterion applied at test time. It would be good to mention this related work in order to highlight that there are further ML tools that could potentially be investigated for such many-to-many mappings.

Authors Response 1:

We appreciate the referee for mentioning this good point. As suggested, we have studied the CycleGAN and found that it can also be potentially used as further ML tool for such many-to-many mappings. For CycleGAN, there are two generator networks and two discriminator networks. The concept of "cycle consistency" comes from the fact that the model includes a cycle-consistency loss, which ensures that, if an image is translated from one domain to another and then back again, it should ultimately look similar to the original image. It helps to avoid generating unrealistic or inconsistent images. To some extent, the mechanism is methodologically similar to our bifurcation tree in the inference phase, as both of them transfer the data from one domain to another, and then back; they both calculate the difference between the original and cycle-transferred data in one domain (i.e., cycle-consistency loss), as shown in Fig. R1.

Fig. R1 | The working mechanism of CycleGAN and cycle consistency. G and F are the abbreviations of two generators, and D_Y and D_X are the abbreviations of two discriminators. The cycle-consistency loss calculates the difference between the original and cycle-transferred data in one domain. The figure is from Ref. [ICCV 2223-2232 (2017)].

Compared with our bifurcation tree selection, the “cycle consistency” introduced in CycleGAN is designed for different technical purpose. As an upgraded version of the pix2pix model [CVPR 1125-1134 (2017)], CycleGAN eliminates the need for paired training data and works as a unsupervised model. In this regard, the incorporation of “cycle consistency” becomes essential to preserve content consistency when performing image style transfer. Besides, CycleGAN has found practicality in applications where data from each domain has inherent shared characteristics/style, such as object transfiguration and season transfer. It realizes the macroscopic and high-level feature mappings between two domains. By contrast, we adopt the idea of "cycle consistency" (i.e., Euclidean distance) to screen out the optimal candidate in our spectra correlation, achieving the precise matching of data pairs at a microscopic level of many-to-many mappings.

In the new submission, we have highlighted the CycleGAN model with its cycle consistency criterion and added the above discussion on lines 234-240,

“The procedure is somehow similar to the “cycle consistency” criterion in the unsupervised model CycleGAN⁴⁵, which also minimizes the distance between the original data and the cycle-transferred data in order to preserve content consistency while performing image style transfer. It requires the data from each domain to possess inherent shared characteristics (i.e., style). By contrast, we use “cycle consistency” for the precise matching of data pairs at a microscopic level of complex many-to-many mapping. The value of the minimal distance will also indicate the rationality of the obtained optimal solution for any random input.”

Referee #1 -- Comment 2:

The results section currently discusses the VAE/ELBO extensively. However, since the VAE/ELBO is an existing method, I think that such discussion would fit better in the methods section at the end of the paper. I would expect the results section to be more focused on the way the two VAEs are used together for the purpose of the application, in particular, highlighting the interconnection between the two VAEs and the bifurcation tree (cycle consistency) criterion.

Authors Response 2:

Thanks for the good suggestion. According to the referee's comments, we have simplified the description about the VAE/ELBO and included it in Methods and Supplementary Materials. Meanwhile, we also highlight the interconnection between two VAEs and the bifurcation tree in the result section; see below.

On lines 113-117, "When two cascaded networks have been trained independently, their decoders will be combined by a hierarchical bifurcating tree. As shown in Fig. 2b, the bidirectional mapping and two-way selection are built up between low-frequency and high-frequency spaces, and the inferior candidates can be excluded by comparing the Euclidean distance between the label input and secondary candidates in the low-frequency space."

On lines 226-228, "Furthermore, we exploit a tree diagram to clarify how we combine the generation "magnifier" and elimination "magnifier" to build up the bidirectional mapping between two spectra spaces in Fig. 2b and obtain the ideal solution."

On lines 230-234, "The core is that the father node with the most orthodox leaf node will be chosen as the final output, and "the most orthodox" is defined as the minimum distance between the root node and the leaf node, or alternately say, the minimum Euclidean distance between the input and secondary candidates in Fig. 2b."

Referee #1 -- Comment 3:

Furthermore, the clarity of the technical section could be improved. Currently, figures do not include the notation and terms used in the paper, and therefore, they do not facilitate the understanding of the technical sections of the paper. Some of the figures are unintuitive, for example, it took me time to realize in Fig. 2a (top) that the blue box is what we actually feed as input, and that the orange boxes at the input and output is some kind of recurrent connection. Maybe a more abstract subfigure showing in a very simple way how the two VAEs interconnect would be useful. Actually, I like Fig. R1 from the authors' response letter, and think it could potentially be used for the main paper.

Authors Response 3:

We thank the referee for the careful reading and the constructive suggestion. As suggested, we have included the notations and terms in Fig. 2a and Fig. 2c to facilitate the understanding. Besides, we have incorporated Fig. R1 from the last response letter as Fig. 2b in the main text, and added the corresponding description to highlight the interconnection between two VAEs; see below.

On lines 125-127, "Meanwhile, the "input" we mentioned in Fig. 1 corresponds to the "Label" (y) in both Fig. 2a and Fig. 2c. It serves as another input element as indicated by the blue boxes."

On line 164, "and then the variables are randomly sampled from the selected Gaussian distribution and decoded into candidates, constituting the solution domain in Fig. 2b."

On lines 444-448, "b, The interconnection between two networks. For a given input (low-frequency), the generation network can generate various candidates (high-frequency), and the elimination network will reversely map each candidate into the original space. The optimal candidate is picked out by calculating the Euclidean distance between the input and secondary candidates."

Referee #1 -- Comment 4:

While the authors have added a quantified comparison, I think the results section remains a bit thin, in particular, it would be good to have at least two distinct data generation / numerical simulation scenarios in order to demonstrate that the high reported performance is reproducible under slightly varying conditions.

Authors Response 4:

Thanks for the good comment. To verify the effectiveness of our method, we further demonstrate two distinct data generation/scenarios. The first case is that we use different metasurface geometries. Figure R2 demonstrates the results of arc and distorted h-shape metasurface patterns, respectively. The first plot in each row is the input spectra from the test dataset, and the second plot is the diverse candidates generated by the generation network. The last plot in each row is the final solution singled out by the elimination network, where the solid line and inset are the ground-truth optical response and design pattern, respectively. The close match between the dashed line and solid line in all cases proves the generality of our model.

Fig. R2 | The results of arc and distorted h-shape metasurface patterns, respectively. a, The input spectra samples from the arc and distorted h-shape testing dataset. **b,** The diverse candidates outputted by the generation network. **c,** Final solutions singled out by the elimination network.

The second case is the global far-field customization of intelligent metasurfaces, for the application of fifth-generation (5G) wireless communication. Intelligent metasurfaces have been found to be a superior candidate for managing wireless channels in a green and cost-effective manner [Sci. Adv. 8, eabn7905 (2022)]. As shown in Fig. R3a, the goal is to steer the main lobe towards the user's direction. For a given direction, our generation-elimination framework will firstly generate various nominated metasurface distributions, and then filter out the optimal one with the criterion of minimal distance regarding the main lobe. As shown in Fig. R3b, the blue curve in each test instance is the input/desired main lobe, and the orange curve is the simulation result of the optimal metasurface design filtered by the elimination network. In both cases, the close match between the orange curve and the blue one proves the versatility of our framework.

Fig. R3 | Demonstration of global far-field customization of wireless communication. **a**, In the wireless communication scene, to steer the main lobe towards the concerned or user's direction, we need to generate various candidates of metasurface distributions and retain the best one. **b**, The results outputted by the framework. In both cases, the blue curve is the input/desired far-field with only a main lobe, and the orange curve is the simulation result of the optimal metasurface design.

In the new submission, we have added the above results in Supplementary Note 5, and noted them in the main text on lines 257-258,

“See Supplementary Note 5 for results on two additional datasets. All of these guarantee the generality and versatility of the proposed framework.”

General comments from Referee #2:

All I concerned in my last comments have been addressed in this revised manuscript, which is quite satisfying. I recommend acceptance of the revised version.

Authors Response:

We thank the referee for the positive comments and the recommendation of our work.